# Constitutive polysaccharide degradation and diet-dependent lipid metabolism reveal an adaptive feeding strategy in the pacific oyster *Magallana gigas*

Manabu W.L. Tanimura[1,2]*, Kazuhiko Koike[3], Kazumi Matsuoka[2,4]

1 Graduate School of Human and Environmental Studies, Kyoto University, Kyoto, Japan, 2 Seed Bank Co. Ltd. Sakyo, Kyoto, Japan, 3 Graduate School of Biosphere Science, Hiroshima University, Hiroshima, Japan, 4 C/O Institute for East China Sea Research, Nagasaki University, Nagasaki, Japan

* manabu.w.l.tanimura@gmail.com

## Abstract

Previous studies have identified several endogenous cellulases in the Pacific oyster *Magallana gigas*, indicating its potential to assimilate terrestrial carbon sources. However, the mere presence of endogenous cellulases suggests capability but does not confirm actual feeding behavior. Here, we analyzed the transcriptomes of two groups of *M. gigas* fed with either cultured diatoms or terrestrial leaves. The results showed that although one GHF9 endoglucanase were upregulated in response to leaf feeding, other cellulase GH families exhibited no significant differences compared to the diatom-fed group. Moreover, xylanase expression levels remained unchanged, suggesting that enzymes involved in the decomposition of terrestrial carbohydrates may be maintained at a stable baseline. In contrast, enzymes related to fatty acid assimilation, specifically β-oxidation enzymes, were upregulated when oysters were fed diatoms. These findings suggest that *M. gigas* maintains a constitutive level of glycoside hydrolase expression to assimilate the "ubiquitous" terrestrial carbon source, while it upregulates related enzymes in response to the "occasional" availability of fatty acids from phytoplankton.

## Introduction

The Pacific oyster *Magallana gigas* (Thunberg, 1793) is currently recognized as the most commercially successful oyster species worldwide, supporting major aquaculture industries across the globe [1]. Traditionally, *M. gigas* has been described as a suspension feeder that consumes a wide variety of benthic and planktonic microalgae, including diatoms and dinoflagellates [2]. These microalgae are rich in essential lipids and fatty acids, which have long been regarded as the primary nutritional sources required for the survival, growth, and reproduction of both larval and adult oysters [3].

**Data availability statement:** Raw sequencing reads generated prior to transcriptome assembly are available in the NCBI Sequence Read Archive (SRA) under accession number PRJNA1362950. Other data analysis files, including annotations of extracted enzymes, results of *M. gigas* genome alignment, transcript quantification analyses, and RSEM output files, are provided in the Appendix.

**Funding:** This study was supported by Grants-in-Aid for Scientific Research from the Ministry of Education, Culture, Sports, Science and Technology of Japan (Grant No. 20KK0141; recipient: Kazuhiko Koike). The funders had no role in study design, data collection and analysis, decision to publish, or preparation of the manuscript.

**Competing interests:** There are no conflicts of interest to declare.

However, several ecological observations have revealed that *M. gigas* can also thrive in turbid and nutrient-poor environments such as mangrove estuaries, where water turbidity limits light penetration and consequently reduces phytoplankton production [4]. The persistence of oysters under such low-phototrophic conditions suggests the presence of alternative feeding strategies or carbon assimilation pathways beyond microalgal consumption.

In our previous research, we identified genes encoding glycoside hydrolases (GHs), specifically cellulase—an enzyme capable of cleaving β-1,4-glycosidic linkages [4]. The presence of this cellulase gene within the *M. gigas* genome strongly indicates the potential ability of this species to utilize terrestrial, plant-derived polysaccharides as supplementary carbon and energy sources. This discovery aligns with a growing body of evidence suggesting that many invertebrates possess endogenous glycoside hydrolases, which enable them to degrade complex organic matter of both terrestrial and marine origin [5–7]. Such enzymatic versatility may play an important role not only in the nutritional ecology of oysters but also in broader biogeochemical processes, contributing to the global carbon cycle.

If *M. gigas* is indeed capable of utilizing cellulose as an alternative carbon source, it is reasonable to hypothesize that the expression of its cellulase genes would be upregulated when cellulose is provided as the sole food source following a period of starvation. Cellulose degradation yields glucose, which serves as a universal and essential energy substrate for nearly all living organisms. However, glucose provides a comparatively lower nutritional value than macromolecules such as proteins or lipids, which offer both higher energy density and essential biochemical components for growth and reproduction.

Based on this understanding, we hypothesize that starved *M. gigas* individuals, when subsequently fed with lipid-rich microalgae such as diatoms (used in the present study), would preferentially assimilate these algal-derived nutrients. Because diatoms contain substantial amounts of lipids but lack cellulose [8], it is expected that *M. gigas* would upregulate genes and enzymes associated with lipid catabolism—particularly those involved in β-oxidation—rather than those linked to cellulose degradation.

This study was therefore designed to test whether *M. gigas* differentially regulates metabolic pathways related to carbohydrate and lipid utilization in response to distinct dietary conditions. The findings are expected to provide new insights into the feeding strategy and metabolic flexibility of *M. gigas*, with potential implications for improving nutritional management and productivity in oyster aquaculture.

## Materials and methods

### *Magallana gigas* sampling and culturing

Wild *M. gigas* specimens were collected from the Onosato River Estuary, Osaka Prefecture, Japan (34°22′35.8″N, 135°15′02.3″E) on January 14, 2024. No permits were required for collecting oysters at this public site, as stated by the Osaka Prefecture Government in Japan. Also, these attached individuals are wild grown and do not belong to any interest groups. The oysters were transported to Kyoto University,

Kyoto Prefecture, Japan, in local seawater. Upon arrival at the laboratory, groups of five individuals measuring more than 5 cm along the major axis were transferred to white 20 L buckets containing 10 L of artificial seawater and aerated at a rate of 5 L min$^{-1}$. Each group of *M. gigas* was first starved for one week. Subsequently, one group was fed daily with 0.5 g of dried reed (*Phragmites australis*) leaves, while the other group received 50 mL of *Chaetoceros gracilis* culture ($1 \times 10^6$ cells mL$^{-1}$) for one week. The survival of all individuals was monitored daily by checking shell closure in response to tactile stimulation. All oysters survived until the end of the experiment.

## Total RNA extraction and purification

Three individuals of *M. gigas* were collected from each experimental group. Digestive glands were excised immediately after the feeding experiment, rinsed sequentially with approximately 25 mL of artificial seawater and 25 mL of distilled water, blotted dry with paper towels, and surface-sterilized with 70% ethanol. The glands were then homogenized using sterilized scissors, and approximately 0.2 mL of homogenate was transferred into 1 mL of TRIzol reagent (Thermo Fisher Scientific, Waltham, MA, USA). Total RNA was extracted following the manufacturer's standard protocol.

## Trimming and De novo assembly

Extracted RNA was sequenced by an external company (Rhelixa, Inc., Tokyo, Japan). Briefly, electrophoresis was performed to assess nucleic acid integrity and concentration prior to library preparation. Poly(A) strand-specific library preparation (dUTP method) was conducted using the NEBNext Poly(A) mRNA Magnetic Isolation Module and the NEBNext Ultra™ II Directional RNA Library Prep Kit (New England Biolabs Japan Inc., Tokyo, Japan).

The resulting double-stranded DNA libraries were sequenced on an Illumina NovaSeq 6000 platform, generating 150 bp paired-end reads with an approximate depth of 10 gigabases (66.7 million reads) per sample. Raw sequencing data were processed using Atria software (version 3.2.1) to trim adapter sequences. The trimmed reads were then assessed for quality using FastQC software (version 3) before assembly with Trinity software (version 2.15.0) using all replicates in two feeding groups to generate an integrated assembly for easy downstream quantification analysis. The overall pipeline of in silico analysis was shown in Fig 1. The completeness and quality of the assembled transcripts were evaluated with BUSCO software (version 5.4.5), referencing the *eukaryote_odb10* dataset.

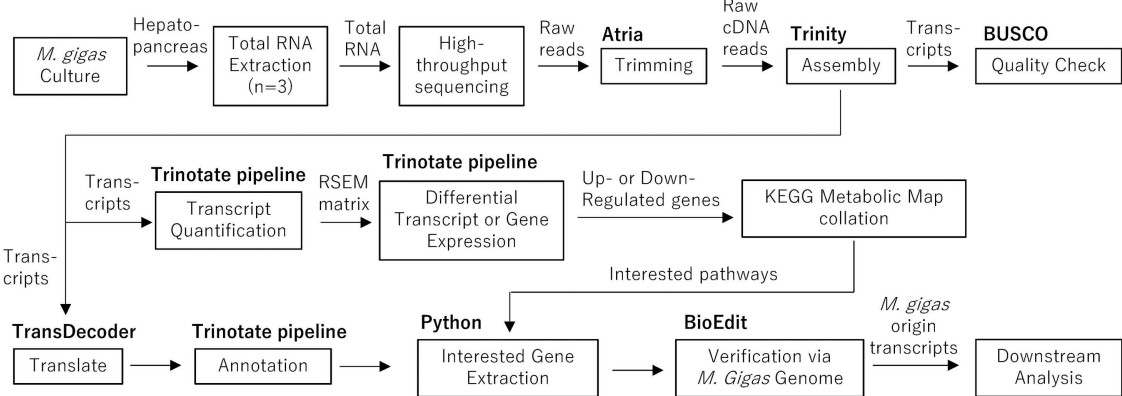

**Fig 1. In silico analysis pipeline used in this study.** Squares represent analysis steps, while bold text denotes the software or tools used. Arrows indicate both the data/material type and the direction of workflow.

## Transcript quantification, differential expression analysis and principal component analysis

Transcript quantification was performed by aligning raw reads to the assembled transcripts using the RSEM method. Transcript abundance was expressed as transcripts per million (TPM). The resulting expression data were analyzed with DESeq2 (version 1.34.0), a package integrated into the Trinity pipeline, to identify differentially expressed transcripts. The output matrix file was subsequently used for downstream analyses as described below. In parallel, the count matrix was analyzed using pcaExplorer (version 3.4.0) to perform principal component analysis (PCA) between the two oyster feeding groups. The isoform-level count matrix was used as input to construct a DESeqDataSet (DDS), from which a DESeqTransform object was generated using regularized logarithm (rlog) transformation. Sample-to-sample correlations were calculated using the Pearson method, and the number of most variable genes included in the analysis was set to 1,500. Biological replicates from each group (diatom or leaf) were visualized in a two-dimensional PCA plot, where the x- and y-axes represent the first and second principal components of gene expression variance, respectively.

## Kyoto Encyclopedia of Genes and Genomes (KEGG) pathway analysis

Differentially expressed transcripts ($p < 0.05$) were uploaded to the KEGG Automatic Annotation Server (KAAS) to identify associated metabolic pathways. The analysis was conducted using the single-directional best-hit method based on the BLAST algorithm (nucleotide sequence mode), referencing major invertebrate gene datasets including Crustacea, Chelicerata, Nematoda, Annelida, Mollusca, Cnidaria, and Porifera. The annotated results were subsequently imported into KEGG Mapper to visualize and identify related pathways and enzymes.

## Functional annotation

Functional annotation was performed by applying Trinotate software (version 3.2.2) pipeline against the UniprotKB/Swiss-Prot, Pfam and Blast2GO database, under default settings.

## Regulated gene extraction

Based on the results of the KEGG pathway analysis and our hypothesis, enzymes related to cellulase, xylanase, pectinase, lipase, and β-oxidation pathways were targeted in this study. Alternative enzyme names were obtained from the Enzyme Nomenclature Database and used as search keywords. The search process was automated using an in-house Python script, which queried the Trinity-annotated transcriptome dataset.

## *M. gigas* genome alignment for contamination filtration

Extracted transcripts corresponding to the targeted enzymes were analyzed using BioEdit (version 7.2.5) with a local BLASTn algorithm (version 2.2.10; *e*-value threshold $= 1 \times 10^{-5}$). The query sequences were aligned against the assembled *M. gigas* genome (accession number GCA_963853765.1) downloaded from the NCBI Genome Database. Transcripts exhibiting sequence similarity above the set threshold were regarded as of *M. gigas* origin, whereas those below the threshold were considered potential contaminants.

## Volcano plot of targeted genes

TPM values and related transcript data were compiled and visualized using Prism software (version 10.6.1). Transcripts identified as potential contaminants were plotted as green dots, non-significant transcripts as black dots, and significantly regulated transcripts ($p < 0.05$) as red dots in the volcano plot.

## Results

### Total RNA extraction, sequencing, trimming and assembly

RNA extracted from *M. gigas* digestive glands had RNA Integrity Number (RIN) values greater than 7.9, with sufficient concentrations for sequencing (S1 Fig). The raw sequencing results generated by the high-throughput sequencer are summarized in S1 Table. All raw sequencing data have been deposited in the National Center for Biotechnology Information (NCBI) database under the accession number PRJNA1362950. Phred quality scores (Q scores) after trimming are shown in S2 Fig, indicating sufficient quality for assembly. Trinity successfully assembled 806,064 transcripts in total. BUSCO analysis evaluated 255 ortholog groups, yielding 92% complete BUSCOs, 1.6% fragmented BUSCOs, and 0.4% missing BUSCOs, demonstrating a high-quality assembly sufficient for downstream analysis.

### Statistical reliability and PCA analysis for data overview

The dispersion plot is shown in S3 Fig. The fitted dispersion values exhibited a smooth mean–dispersion trend, and the final dispersion estimates followed this trend quite closely, indicating that variance was appropriately modeled across transcripts and supporting the statistical reliability of the differential expression analysis.

Prior to KEGG mapping, principal component analysis (PCA) was performed on isoform-level transcript counts (Fig 2). The two-dimensional plot, defined by PC1 and PC2, explained approximately 42.3% of the total variance, indicating that differences between the two feeding groups did not result in clearly distinct transcriptomic profiles (Fig 2).

### Regulated metabolic pathways

Mapping the up- and down-regulated genes identified through the Trinity Differential Expression Analysis pipeline onto KEGG metabolic pathways revealed four major pathways associated with polysaccharide and lipid degradation (Fig 3).

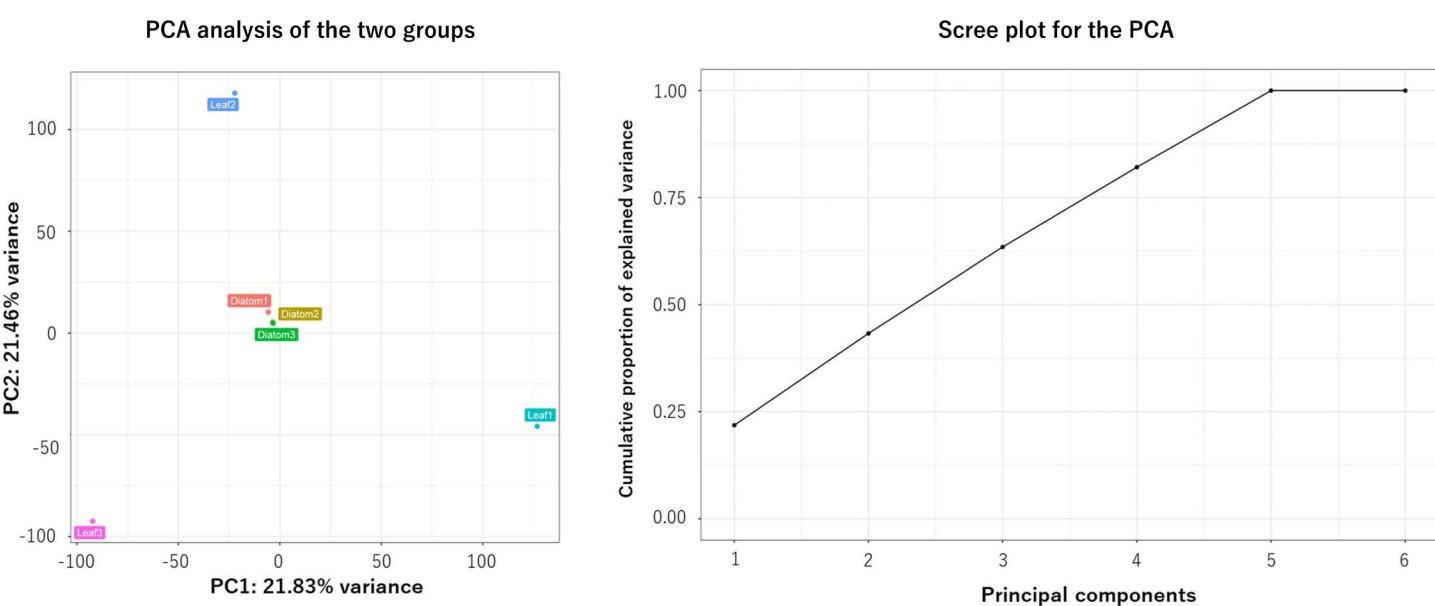

**Fig 2. Principal component analysis (PCA) of isoform-level transcript expression.** Left, PCA plot based on isoform-level transcript counts, showing the distribution of biological replicates from the diatom-fed and reed leaf–fed groups. Each point represents one biological replicate. Samples from the diatom-fed group cluster closely together, whereas those from the reed leaf–fed group are more widely dispersed, indicating greater variability in transcriptomic responses under the reed leaf feeding condition. Right, Scree plot showing the proportion of variance explained by each principal component. The first two principal components (PC1 and PC2) together explained 42.3% of the total variance.

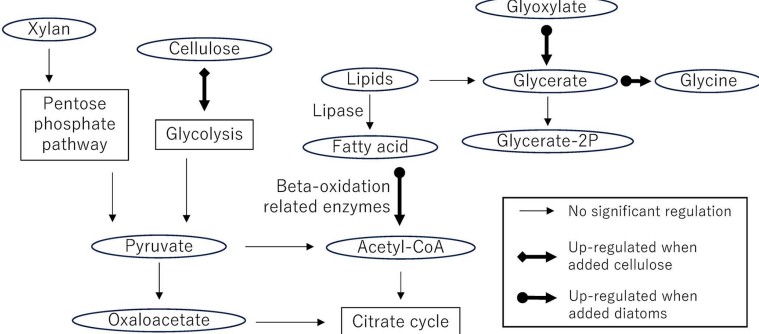

**Fig 3. Metabolic pathways of interest in this study, including glycolysis, the pentose phosphate pathway, and lipid catabolism.** Thin arrows indicate the detection of corresponding enzymes in *M. gigas*, while bold arrows represent upregulated enzymes under either the cellulose-fed or diatom-fed conditions.

Because the KEGG Mapper tool does not allow direct specification of differential expression groups, an additional volcano plot was constructed using transcripts extracted from the Trinotate annotation pipeline, in which enzyme commission (EC) numbers were employed as search keywords (Table 1, S1 Appendix).

In total, six cellulase genes comprising 36 isoforms were extracted from the transcriptome, along with 11 xylanase genes (42 isoforms), 103 lipase genes (453 isoforms), and 87 β-oxidation–related enzyme genes (359 isoforms). In addition, no pectinase genes/isoforms were detected in the total transcripts. During the *M. gigas* genome alignment step, all

**Table 1. Summary of the extracted enzymes of interest. Total counts of both genes and isoforms are presented. For each of the four enzyme types, the numbers of genes and transcripts aligned with the *M. gigas* genome and those unaligned (presumed contaminants) are shown.**

| Total Transcripts | Isoforms | | Genes | |
|---|---|---|---|---|
| | 719,074 | | 408,401 | |
| **Cellulase** | Isoforms | | Genes | |
| | 36 | | 7 | |
| | **M.gigas genome aligned** | **Potential eukaryotic contamination (not aligned to M. gigas genome)** | **M.gigas genome aligned** | **Potential eukaryotic contamination (not aligned to M. gigas genome)** |
| | 36 | 0 | 7 | 0 |
| **Xylanase** | Isoforms | | Genes | |
| | 42 | | 11 | |
| | **M.gigas genome aligned** | **Potential eukaryotic contamination (not aligned to M. gigas genome)** | **M.gigas genome aligned** | **Potential eukaryotic contamination (not aligned to M. gigas genome)** |
| | 42 | 0 | 11 | 0 |
| **Lipase** | Isoforms | | Genes | |
| | 453 | | 103 | |
| | **M.gigas genome aligned** | **Potential eukaryotic contamination (not aligned to M. gigas genome)** | **M.gigas genome aligned** | **Potential eukaryotic contamination (not aligned to M. gigas genome)** |
| | 441 | 12 | 99 | 4 |
| **beta-oxidation related enzymes** | Isoforms | | Genes | |
| | 359 | | 85 | |
| | **M.gigas genome aligned** | **Potential eukaryotic contamination (not aligned to M. gigas genome)** | **M.gigas genome aligned** | **Potential eukaryotic contamination (not aligned to M. gigas genome)** |
| | 347 | 12 | 81 | 4 |

cellulase- and xylanase-related transcripts showed clear matches to the *M. gigas* genome, confirming their endogenous origin. In contrast, four lipase genes (12 isoforms) and four β-oxidation–related genes (12 isoforms) did not align with the *M. gigas* reference genome and were therefore considered potential contaminants derived from other species.

All extracted isoforms, including these potential contaminants, were visualized in the volcano plot (Fig 4, S2–S3 Appendix), with contaminant transcripts represented in a distinct color. The resulting patterns indicated that cellulase-related transcripts exhibited little differential regulation between the two dietary treatments, except for a single transcript that was significantly upregulated in the cellulose-fed group. Xylanase genes displayed a similar trend, with most transcripts showing no significant changes in expression, except for one transcript that was modestly upregulated in the diatom-fed group with a *p*-value slightly exceeding the 0.05 threshold. It should be noted that the p-values used in the present study were not corrected for the false discovery rate (FDR). Although the application of FDR correction reduced the number of significantly differentially expressed transcripts, thereby minimizing false positives arising by chance, it did not alter the overall interpretation of the study. Furthermore, as the primary aim of this work is to explore general metabolic response patterns rather than to define a strict list of significantly regulated transcripts, FDR-corrected q-values were not applied in the main analysis.

By contrast, lipase-related transcripts displayed more complex regulation, with numerous isoforms showing both up- and down-regulation across treatments. The cellulose-fed oysters tended to exhibit a slightly higher number of upregulated lipase transcripts with greater fold changes compared to the diatom-fed group. Enzymes associated with β-oxidation also demonstrated multiple regulatory shifts; however, in this case, the diatom-fed group showed a markedly higher number of upregulated transcripts and greater fold changes, indicating an enhanced activation of lipid catabolic pathways under lipid-rich dietary conditions.

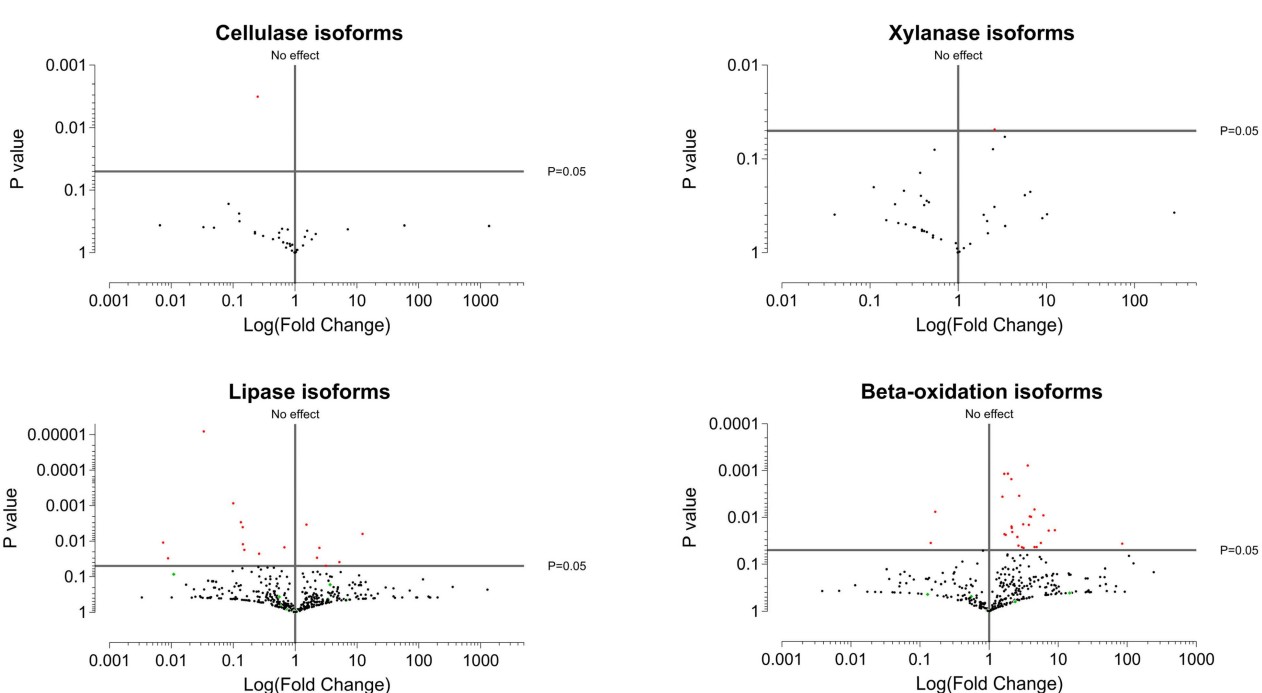

**Fig 4. Volcano plots of the four enzyme groups of interest.** The horizontal axis represents the log₂ fold change in expression. Transcripts to the left of the "no effect" line are upregulated in the cellulose-fed group, whereas those to the right are upregulated in the diatom-fed group. The vertical axis represents *p* values, with a reference line at *p* = 0.05 for visualization.

## Discussion

### Global transcriptomic variation assessed by PCA

As shown in Fig 2, the three diatom-fed replicates clustered closely together, whereas the three leaf-fed replicates were more widely dispersed across the plot. This pattern suggests that *M. gigas* exhibits a more consistent metabolic response when fed with diatoms, while its response to reed leaves is more variable among individuals. One possible explanation is that reed leaves do not induce a specific or coordinated metabolic response in *M. gigas*. However, this interpretation contrasts with our previous findings that *M. gigas* possesses endogenous glycoside hydrolases capable of degrading terrestrial plant polysaccharides. Alternatively, this observation may indicate that these enzymes are not actively regulated in the presence of reed leaves, possibly because processing such substrates does not require a distinct metabolic adjustment.

### Regulation of polysaccharide hydrolysis related enzymes

The commonly held assumption that gene expression levels are "upregulated when needed" does not appear to apply straightforwardly to *M. gigas* cellulases. In the present study, the expression levels of cellulase genes in *M. gigas* did not differ significantly between individuals fed with cellulose and those fed with diatoms. Nevertheless, substantial inter-individual variability in cellulase expression was observed among the oysters analyzed, regardless of feeding condition. Several explanations may account for this phenomenon.

First, it is possible that *M. gigas* maintains a constitutive level of cellulase expression because cellulose is one of the most abundant organic compounds on Earth and is expected to accumulate in estuarine environments where this species typically inhabits [9]. In this scenario, the oysters may continuously express cellulase to assimilate cellulose as a routine carbon source. Although glucose—the monomer produced from cellulose hydrolysis—has relatively low nutritional value compared with lipids or proteins, it remains an essential substrate for cellular respiration and muscle energy metabolism. Thus, *M. gigas* might sustain a baseline level of cellulase expression to secure a minimal but steady supply of glucose for energy production, rather than dynamically adjusting its expression according to food availability.

Second, the cellulases possessed by *M. gigas* may not function exclusively in degrading free cellulose from terrestrial sources but could also serve to digest algal cell walls. Many green algae and dinoflagellates possess cellulose or cellulose-like components in their thecal plates or cell walls [8]. Therefore, cellulase activity in *M. gigas* may facilitate the mechanical or enzymatic breakdown of these microalgae, which are known components of the oyster's natural diet [2]. In support of this interpretation, the only cellulase isoform that was upregulated in the cellulose-fed group was identified as an endoglucanase (predicted by AlphaFold3; Average pLDDT = 95.25) [10]. Endoglucanases cleave internal β-1,4 linkages of cellulose chains, producing shorter oligosaccharides rather than glucose monomers [7,9]. Such an enzyme would be more effective for loosening structural cellulose within algal cell walls than for completely digesting free cellulose, consistent with previous reports indicating that *M. gigas* commonly feeds on green algae and dinoflagellates.

Third, the one-week starvation period prior to feeding might have been insufficient to induce a measurable physiological response. Bivalves are known to tolerate prolonged starvation by reducing metabolic activity and even catabolizing their own muscle tissues [11,12]. Therefore, a longer deprivation period might be required for *M. gigas* to perceive starvation as a stress condition that necessitates the upregulation of cellulase expression for emergency glucose production.

Despite the absence of significant upregulation under the present experimental conditions, the consistent expression of cellulase genes suggests that *M. gigas* maintains a constitutive enzymatic capacity to assimilate cellulose regardless of dietary composition. This persistent baseline activity may partly explain its ecological success in turbid estuarine environments, where phytoplankton productivity is often limited but detrital cellulose from terrestrial sources is abundant.

However, alternative explanations such as post-transcriptional regulation or slow transcriptional kinetics are also plausible. Future studies should include longer-term mesocosm feeding experiments and quantify total organic carbon and particulate matter in natural environments to better elucidate how *M. gigas* balances its carbon and energy budgets.

Such analyses will also help clarify the functional significance of possessing cellulase—whether it primarily serves cellulose assimilation, algal cell-wall degradation, or other physiological roles. Stable isotope tracing could further determine whether cellulose-derived carbon is truly incorporated into oyster biomass. Nevertheless, isotopic approaches must be interpreted with caution. As demonstrated in our previous gastropod feeding experiment, even when cellulose ingestion was visually confirmed, isotopic signatures in tissues remained largely unchanged [5]. This observation suggests that much of the ingested cellulose may be oxidized through respiration, producing $CO_2$ that is released into the environment rather than retained in tissues, thereby leaving no measurable isotopic signal within the organism.

A similar pattern was observed for xylanase expression: no significant differences were detected between the cellulose-fed and diatom-fed groups. Xylan, a major hemicellulosic polysaccharide predominantly found in terrestrial plant cell walls, contributes to cell wall rigidity by cross-linking with cellulose and lignin [13]. Many invertebrates are known to possess endogenous xylanases, suggesting a widespread ability to degrade xylan and, by extension, participate in the turnover of plant-derived detritus. The enzymatic breakdown of xylan can facilitate cellulose degradation by loosening the cell wall matrix, thereby increasing cellulase accessibility to cellulose fibers [14]. In addition, xylose, the monosaccharide product of xylan hydrolysis, can enter the pentose phosphate pathway, providing reducing power and metabolic intermediates for biosynthesis and energy metabolism [15–17].

However, the ecological and physiological roles of xylanase in most invertebrates remain poorly understood. In the present study, *M. gigas* exhibited relatively stable xylanase expression levels under both dietary treatments, suggesting that these enzymes are constitutively expressed rather than inducibly regulated. Such a stable expression pattern may indicate a general capacity to degrade xylan as part of the species' routine metabolic repertoire, rather than a direct transcriptional response to dietary xylan, just like cellulase. Previous studies have shown that certain red and green algae possess xylose residues within their cell wall polysaccharides [18,19]. In contrast, *Chaetoceros*, the diatom species used as the food source in this experiment, does not contain significant content of xylan or xylan-like structures in its cell wall, rather than alpha-1,3-mannan [20,21]. At present, there is also no clear evidence that the microalgae commonly consumed by *M. gigas* in natural environments possess such xylan components [8]. Therefore, the precise functional significance of xylanase activity in *M. gigas* remains speculative and warrants further biochemical and ecological investigation.

We did not detect any pectinase-related transcripts in the present study. This was somewhat surprising to us, because the presence of cellulases and xylanases suggests that terrestrial plant–derived polysaccharides are indeed part of the dietary spectrum of *M. gigas*, which would intuitively make the possession of pectinase seem likely as well. However, no pectinase-related transcripts were identified in the present dataset, and searches of the NCBI database likewise did not return any clear pectinase hits for *Magallana gigas*. We propose several possible explanations for this result. First, *M. gigas* may have lost or never retained pectinase genes during its evolutionary history, possibly because the ability to degrade pectin does not confer a strong nutritional advantage compared with cellulose degradation, and the breakdown products of pectin may not play a major role in its metabolic pathways. Second, pectin may not be a major component of particulate organic matter (POM) available to oysters in coastal environments. Unfortunately, quantitative studies examining the relative abundance of different polysaccharides in seawater and sediment remain limited, making this hypothesis difficult to evaluate at present. Nevertheless, the apparent absence of pectinase in oysters is intriguing from both physiological and ecological perspectives and warrants further investigation in future studies.

## Regulation of lipid lysis related enzymes

Lipases are a broad class of enzymes that catalyze the hydrolysis of lipids into free fatty acids and glycerol, thereby facilitating lipid mobilization and energy production [22]. Because diatoms are well known to be rich in lipids [23], it would be reasonable to hypothesize that, after a period of starvation, *M. gigas* would upregulate lipase expression to digest and utilize diatom-derived lipids. However, the results of this study revealed a somewhat opposite trend. Although lipase-related transcripts in *M. gigas* exhibited a higher degree of differential regulation (i.e., more isoforms showing fold changes

with $p < 0.05$) than cellulase or xylanase, the oysters fed with cellulose showed a greater number of upregulated lipase isoforms than those fed with diatoms. Previous studies have reported that *M. gigas* tends to accumulate lipid reserves when food is abundant and to deplete these reserves during periods of food scarcity [24]. Similarly, Su [25] demonstrated that the dynamics of lipid storage and depletion in oysters depend strongly on both the quantity and the nutritional quality of available food. In this context, the observed lipase upregulation in the cellulose-fed group can be interpreted as a metabolic response to nutritional stress. Because cellulose provides relatively low caloric and nutrient value compared with protein- or lipid-rich diets, the oysters likely mobilized endogenous lipid stores to meet their energetic demands, leading to increased lipase expression for lipid hydrolysis. In contrast, oysters fed with diatoms did not exhibit strong lipase upregulation. This may indicate that, rather than catabolizing lipids immediately, *M. gigas* preferentially assimilated and stored diatom-derived lipids for future use. Such a strategy would allow the oysters to accumulate energy reserves during periods of high food availability, which could later be mobilized during starvation, reproduction, or environmental stress. Taken together, these results suggest that lipase regulation in *M. gigas* reflects a balance between lipid storage and mobilization, closely linked to the nutritional status and quality of the available diet.

In contrast to the patterns observed for lipase, the enzymes involved in β-oxidation showed a distinctly clear trend: *M. gigas* individuals fed with diatoms exhibited substantially more upregulated isoforms than those fed with cellulose. β-oxidation is a critical metabolic pathway that allows fatty acids to be broken down into acetyl-CoA, which then enters the tricarboxylic acid (TCA) cycle to generate ATP and supply intermediates for multiple downstream biosynthetic pathways [22]. Previous research has demonstrated that the fatty acid profile of *M. gigas* closely mirrors that of the microalgal feed supplied [25], suggesting that oysters accumulate microalgae-derived fatty acids alongside lipid reserves.

The results of the present study suggest that when *M. gigas* are supplied with a lipid- and fatty acid-rich diet (i.e., diatoms), they appear to prioritize the digestion of those fatty acids. This priority is manifested in the upregulation of β-oxidation–related enzymes, likely reflecting an enhanced metabolic routing of fatty acids into energy production and biosynthesis. Conversely, when fed with cellulose—which lacks appreciable fatty acid content—the cellulose-fed group displayed minimal upregulation of β-oxidation enzymes. This observation implies that these enzymes are not maintained at a constitutive high level but are instead induced when fatty acid substrates are abundant.

### Some other possibilities

The diatom-fed group received 50 mL of culture ($1 \times 10^6$ cells $mL^{-1}$) per day, whereas the leaf-fed group received only 0.5 g dry weight of reed leaves. These diets differ substantially in nutritional quality. In addition to the possibilities discussed above, it is also possible that the leaf-fed group experienced a condition close to starvation, which may have influenced their transcriptomic expression patterns. Indeed, the PCA results showed that replicates from the leaf-fed group did not cluster together, suggesting that their transcriptomic responses varied at the individual level. This may indicate that the oysters were responding in a less coordinated manner, potentially exploring multiple metabolic strategies to cope with this low-nutrient condition.

It is also known that bivalves can undergo long-term metabolic suppression. In fact, this phenomenon is utilized in the Japanese fishery industry, where *Corbicula japonica* is subjected to short-term starvation prior to marketing to enhance umami-related compounds such as succinic acid and ornithine. The transcriptomic changes observed in the present study, which involved only one week of starvation, may therefore represent responses to an intermediate or partial starvation state. Future studies should examine different durations of starvation to better understand the metabolic capabilities of *M. gigas* and its responses to varying dietary conditions.

### Conclusion

The nutrient assimilation strategy of *M. gigas* for polysaccharides, lipids, and fatty acids appears to be far more sophisticated than a simple "upregulate when needed" or "digest immediately" mechanism. Although further studies are required

to verify the hypotheses proposed in this work, our findings suggest that *M. gigas* adopts a flexible and adaptive metabolic strategy. Specifically, the oyster seems to utilize abundant terrestrial plant-derived polysaccharides, such as cellulose and xylan, as a routine and steady energy source, while preferentially accumulating lipids when they are available. Moreover, *M. gigas* appears to prioritize the digestion of fatty acids before neutral lipids, enabling efficient energy production when lipid-rich microalgae are accessible.

Such a strategy would provide a clear ecological advantage for a sessile filter feeder like *M. gigas*, whose nutrient intake depends entirely on the availability and composition of suspended particles carried by seawater currents. Because high-value, lipid-rich diets are not consistently available in estuarine environments, maintaining constitutive expression of polysaccharide-degrading enzymes and the ability to store and mobilize lipids when necessary likely contributes to the species' resilience and success across diverse habitats. This metabolic versatility may help explain the exceptional adaptability and global dominance of *M. gigas* in both natural and aquaculture ecosystems.

## Supporting information

**S1 Fig. Quality assessment of extracted total RNA by electrophoresis.** Top row, from left to right: reed leaf–fed group replicates 1–3. Bottom row, from left to right: diatom-fed group replicates 1–3.
(TIF)

**S2 Fig. Quality assessment of assembled transcripts using Phred quality (Q) scores.** Top row, from left to right: reed leaf–fed group replicates 1–3. Bottom row, from left to right: diatom-fed group replicates 1–3.
(TIF)

**S3 Fig. Dispersion estimates from DESeq2.** Each point represents an individual transcript. Black dots indicate gene-wise dispersion estimates, red dots represent the fitted dispersion trend, and blue dots show the final dispersion estimates after shrinkage. The final dispersion estimates (blue) closely follow the fitted trend (red), indicating appropriate modeling of the mean–variance relationship and reliable estimation of dispersion across transcripts.
(TIF)

**S1 Table. Summary of high-throughput sequencing results.**
(XLSX)

**S1 Appendix. Annotation of extracted enzymes, results of *M. gigas* genome alignment, and transcript quantification (TPM) analysis.**
(XLSX)

**S2 Appendix. Results of TPM analysis of total transcript isoforms based on RSEM.**
(TXT)

**S3 Appendix. Results of TPM analysis of total transcript genes based on RSEM.**
(TXT)

## Author contributions

**Conceptualization:** MANABU W.L. TANIMURA.

**Data curation:** MANABU W.L. TANIMURA.

**Formal analysis:** MANABU W.L. TANIMURA.

**Funding acquisition:** Kazuhiko Koike.

**Methodology:** MANABU W.L. TANIMURA.

**Project administration:** Kazumi Matsuoka.

**Resources:** Kazumi Matsuoka.

**Software:** MANABU W.L. TANIMURA.

**Supervision:** Kazuhiko Koike, Kazumi Matsuoka.

**Validation:** MANABU W.L. TANIMURA.

**Visualization:** MANABU W.L. TANIMURA.

**Writing – original draft:** MANABU W.L. TANIMURA.

**Writing – review & editing:** MANABU W.L. TANIMURA, Kazumi Matsuoka.

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
