## [Decision Letter · Decision Letter 0]

5 Mar 2026

PONE-D-25-61177Constitutive Polysaccharide Degradation and Diet-Dependent Lipid Metabolism Reveal an Adaptive Feeding Strategy in the Pacific Oyster Magallana gigasPLOS One

Dear Dr.  TANIMURA,

Thank you for submitting your manuscript to PLOS ONE. After careful consideration, we feel that it has merit but does not fully meet PLOS ONE’s publication criteria as it currently stands. Therefore, we invite you to submit a revised version of the manuscript that addresses the points raised during the review process.

If applicable, we recommend that you deposit your laboratory protocols in protocols.io to enhance the reproducibility of your results. Protocols.io assigns your protocol its own identifier (DOI) so that it can be cited independently in the future. For instructions see: https://journals.plos.org/plosone/s/submission-guidelines#loc-laboratory-protocols. Additionally, PLOS ONE offers an option for publishing peer-reviewed Lab Protocol articles, which describe protocols hosted on protocols.io. Read more information on sharing protocols at . Additionally, PLOS ONE offers an option for publishing peer-reviewed Lab Protocol articles, which describe protocols hosted on protocols.io. Read more information on sharing protocols at https://plos.org/protocols?utm_medium=editorial-email&utm_source=authorletters&utm_campaign=protocols..

We look forward to receiving your revised manuscript.

Kind regards,

Mohammad Moniruzzaman

Academic Editor

PLOS One

Journal Requirements:

“There are no conflicts of interest to declare.”

5. Please note that funding information should not appear in any section or other areas of your manuscript. We will only publish funding information present in the Funding Statement section of the online submission form. Please remove any funding-related text from the manuscript.

6. Please note that your Data Availability Statement is currently missing the repository name and/or the DOI/accession number of each dataset OR a direct link to access each database. If your manuscript is accepted for publication, you will be asked to provide these details on a very short timeline. We therefore suggest that you provide this information now, though we will not hold up the peer review process if you are unable.

7. When completing the data availability statement of the submission form, you indicated that you will make your data available on acceptance. We strongly recommend all authors decide on a data sharing plan before acceptance, as the process can be lengthy and hold up publication timelines. Please note that, though access restrictions are acceptable now, your entire data will need to be made freely accessible if your manuscript is accepted for publication. This policy applies to all data except where public deposition would breach compliance with the protocol approved by your research ethics board. If you are unable to adhere to our open data policy, please kindly revise your statement to explain your reasoning and we will seek the editor's input on an exemption. Please be assured that, once you have provided your new statement, the assessment of your exemption will not hold up the peer review process.

8. We are unable to open your Supporting Information file “S3 Appendix and S4 Appendix” Please kindly revise as necessary and re-upload.

Reviewer's Responses to Questions

**Comments to the Author**

1. Is the manuscript technically sound, and do the data support the conclusions?

Reviewer #1: Yes

Reviewer #2: Yes

2. Has the statistical analysis been performed appropriately and rigorously? 

Reviewer #1: Yes

Reviewer #2: Yes

3. Have the authors made all data underlying the findings in their manuscript fully available?

Reviewer #1: Yes

Reviewer #2: Yes

4. Is the manuscript presented in an intelligible fashion and written in standard English?

Reviewer #1: Yes

Reviewer #2: Yes

5. Review Comments to the Author

Reviewer #1: The manuscript reports the findings of an original research work. The hypothesis was clearly stated and the experiment was conducted logically. The findings are interesting and the limitations of the study are well-stated for guiding the next step. The conclusions are made based on the evidence.

Reviewer #2: Dear Authors,

In this study, it is noteworthy that the transcriptomic responses of Magallana gigas related to polysaccharide degradation and lipid metabolism were examined under two different dietary conditions (terrestrial leaves and diatoms). The hypotheses that cellulase and hemicellulase enzymes may be constitutively expressed, while lipid and β-oxidation pathways may be regulated depending on diet composition, were tested.

In my opinion, the hypothesis is meaningful from both ecological and metabolic perspectives. The experimental design directly aims to test this assumption. In this respect, the study presents a conceptually strong and coherent framework.

The sequencing depth appears balanced between groups (approximately 34–39 million reads per sample, similar %GC values, and no low-quality sequences). The transcriptome assembly quality (BUSCO 92% complete) is at an acceptable level. Performing contamination control through genome alignment is also a positive methodological approach.

The clear up-regulation of enzymes related to β-oxidation in the diatom group is biologically consistent and represents the strongest finding of the study.

However, I suggest some minor improvements:

Adding a PCA plot to show the distribution of replicates would be helpful.

The model formula used in the DESeq2 analysis should be clearly stated.

It should be clarified whether p-values were adjusted using FDR correction.

Briefly mentioning dispersion values would increase statistical reliability.

Regarding diet composition, the cellulose group received 0.5 g dry leaf, while the diatom group received 50 mL (1 × 10⁶ cells mL⁻¹). These two diets do not appear to be equivalent in terms of energy and nutrient content. Therefore, the comparison may reflect not only a “carbon type difference” but also a “nutritional quality difference.” This point is particularly important when interpreting lipase regulation. A short evaluation of the nutritional content of the diets and acknowledgment that the cellulose group may represent a lower-quality diet scenario would make the interpretation more balanced.

It is known that bivalves can perform long-term metabolic suppression. Therefore, it may be briefly discussed whether one week of starvation is sufficient to induce a strong transcriptional response.

The absence of pectinase detection is noteworthy. Since terrestrial leaf tissues may contain pectin, this point could be briefly discussed in the context of the literature, if possible.

Consistency should be ensured in the use of p < 0.01 and p < 0.05 throughout the text.

In addition, some figures in the manuscript file have low resolution. Especially under the heading “Regulated metabolic pathways,” part of the table is not clearly readable. Improving figure and table resolution would enhance the presentation quality.

I find the findings related to lipid metabolism particularly convincing. With the clarifications and minor revisions mentioned above, I believe the study is suitable for publication.

Sincerely,

Referee

6. PLOS authors have the option to publish the peer review history of their article (what does this mean?). If published, this will include your full peer review and any attached files.). If published, this will include your full peer review and any attached files.

.

Reviewer #1: No

Reviewer #2: **Yes:** Ercument GencErcument Genc

---

## [Author Response · Author response to Decision Letter 1]

26 Mar 2026

Editor: All additional requirements from the editor have been addressed.

Reviewer #1: The manuscript reports the findings of an original research work. The hypothesis was clearly stated and the experiment was conducted logically. The findings are interesting and the limitations of the study are well-stated for guiding the next step. The conclusions are made based on the evidence.

ANS: We sincerely appreciate the reviewer’s comments and are pleased that the reviewer agrees with the content of this manuscript.

Reviewer #2: Dear Authors,

In this study, it is noteworthy that the transcriptomic responses of Magallana gigas related to polysaccharide degradation and lipid metabolism were examined under two different dietary conditions (terrestrial leaves and diatoms). The hypotheses that cellulase and hemicellulase enzymes may be constitutively expressed, while lipid and β-oxidation pathways may be regulated depending on diet composition, were tested.

In my opinion, the hypothesis is meaningful from both ecological and metabolic perspectives. The experimental design directly aims to test this assumption. In this respect, the study presents a conceptually strong and coherent framework.

The sequencing depth appears balanced between groups (approximately 34–39 million reads per sample, similar %GC values, and no low-quality sequences). The transcriptome assembly quality (BUSCO 92% complete) is at an acceptable level. Performing contamination control through genome alignment is also a positive methodological approach.

The clear up-regulation of enzymes related to β-oxidation in the diatom group is biologically consistent and represents the strongest finding of the study.

However, I suggest some minor improvements:

Adding a PCA plot to show the distribution of replicates would be helpful.

ANS: Thank you very much for this valuable suggestion.

As this study primarily focuses on enzymes related to polysaccharide hydrolysis and lipid catabolism, we did not initially consider PCA as part of our analytical approach. However, upon performing PCA, we obtained interesting and informative results.

As shown in Fig. 2, M. gigas appears to exhibit a consistent metabolic response when fed with diatoms, as the three replicates of the diatom-fed group cluster closely together. In contrast, the replicates from the reed leaf–fed group are more dispersed, suggesting greater variability in metabolic responses at the individual level. This pattern may indicate that M. gigas adopts multiple metabolic strategies, or simply becomes somewhat uncertain and tries various responses to cope with this low-nutrient diet environment.

Although this does not constitute direct evidence, it makes sense to us in light of the idea that substrates that are ubiquitously available in the environment (e.g., terrestrial polysaccharides) may not trigger substantial metabolic adjustments.

The model formula used in the DESeq2 analysis should be clearly stated.

ANS: the model version of DESeq2 has been clearly stated.

It should be clarified whether p-values were adjusted using FDR correction.

ANS: In the present study, p-values were not initially adjusted using FDR correction. We recognize the potential risk of false positives and therefore performed an additional analysis with FDR correction (desired false discovery rate = 20%) for confirmation. After FDR correction, one cellulase-related transcript remained significant (being “discovery”), one xylanase-related transcript was no longer significant, the number of significant lipase-related transcripts decreased from 17 to 3, and the number of significant β-oxidation–related transcripts decreased from 29 to 4.

Importantly, these changes did not alter the overall interpretation of the study. The purpose of this work is to explore general metabolic response patterns rather than to define a strict list of significantly regulated transcripts. In addition, given the relatively small number of biological replicates, FDR correction may be overly conservative and may mask biologically meaningful trends. For these reasons, we chose not to replace the original figures with the FDR-corrected version, but instead to clarify this point and report the FDR-corrected results in the main text.

Briefly mentioning dispersion values would increase statistical reliability.

ANS: We thank the reviewer for this helpful suggestion. The analysis was conducted using our in-house-built R script, which incorporated functions from the DESeq2 package for differential expression modeling and dispersion estimation, and this has now been briefly described in the revised manuscript. As shown in the dispersion plot, the fitted dispersion values exhibit a smooth mean–dispersion trend, and the final dispersion estimates (blue dots) follow this trend quite closely.

Regarding diet composition, the cellulose group received 0.5 g dry leaf, while the diatom group received 50 mL (1 × 10⁶ cells mL⁻¹). These two diets do not appear to be equivalent in terms of energy and nutrient content. Therefore, the comparison may reflect not only a “carbon type difference” but also a “nutritional quality difference.” This point is particularly important when interpreting lipase regulation. A short evaluation of the nutritional content of the diets and acknowledgment that the cellulose group may represent a lower-quality diet scenario would make the interpretation more balanced.

ANS: We thank the reviewer for this important comment. We agree that the two feeding treatments were not nutritionally equivalent and that the observed differences may reflect not only carbon source type, but also differences in overall nutritional quality and energy availability.

We have now clarified this point in the revised manuscript and discussed the possibility that the transcriptomic responses observed in the reed leaf–fed group, particularly those related to lipid metabolism, may reflect not only substrate preference but also a low-nutrient or near-starvation physiological state.

It is known that bivalves can perform long-term metabolic suppression. Therefore, it may be briefly discussed whether one week of starvation is sufficient to induce a strong transcriptional response.

ANS: We agree that one week of starvation may not necessarily be sufficient to induce the full transcriptional response associated with severe nutrient deprivation.

We have now added discussion in the revised manuscript to acknowledge that the transcriptomic changes observed in the present study may represent responses to an intermediate or partial starvation state rather than to complete metabolic suppression. This may also help explain the variability observed in the reed leaf–fed group. We agree that future studies comparing multiple starvation durations would be valuable for better understanding the timing and extent of transcriptional responses in M. gigas under different dietary conditions.

The absence of pectinase detection is noteworthy. Since terrestrial leaf tissues may contain pectin, this point could be briefly discussed in the context of the literature, if possible.

ANS: We are glad that the reviewer found this point interesting. We also found it both surprising and intriguing that M. gigas does not appear to possess any pectinase-related genes. One possible explanation is that pectin degradation may not provide sufficient nutritional advantage for M. gigas, especially when compared with other substrates such as proteins, lipids, or even cellulose. Another possibility is that pectin may not be sufficiently abundant in the oyster’s natural environment to favor the retention of such genes through evolution. We have now added a paragraph discussing these possibilities in the revised manuscript.

Consistency should be ensured in the use of p < 0.01 and p < 0.05 throughout the text.

ANS: Thank you for this comment. The p-value thresholds have now been revised to ensure consistency throughout the manuscript (the set of transcripts used for the KAAS search was reanalyzed and revised accordingly). This modification did not alter any of the conclusions or results presented in the paper, as this step was used only for the extraction of potentially differentially expressed pathways.

In addition, some figures in the manuscript file have low resolution. Especially under the heading “Regulated metabolic pathways,” part of the table is not clearly readable. Improving figure and table resolution would enhance the presentation quality.

ANS: High-resolution versions of the figures and tables will be provided in the revised manuscript to improve clarity and presentation quality.

I find the findings related to lipid metabolism particularly convincing. With the clarifications and minor revisions mentioned above, I believe the study is suitable for publication.

Sincerely,

Referee

---

## [Decision Letter · Decision Letter 1]

6 Apr 2026

Constitutive Polysaccharide Degradation and Diet-Dependent Lipid Metabolism Reveal an Adaptive Feeding Strategy in the Pacific Oyster Magallana gigas

PONE-D-25-61177R1

Dear Dr. TANIMURA,

We’re pleased to inform you that your manuscript has been judged scientifically suitable for publication and will be formally accepted for publication once it meets all outstanding technical requirements.

An invoice will be generated when your article is formally accepted. Please note, if your institution has a publishing partnership with PLOS and your article meets the relevant criteria, all or part of your publication costs will be covered. Please make sure your user information is up-to-date by logging into Editorial Manager at Editorial Manager® and clicking the ‘Update My Information' link at the top of the page. For questions related to billing, please contact  and clicking the ‘Update My Information' link at the top of the page. For questions related to billing, please contact billing support..

Kind regards,

Mohammad Moniruzzaman

Academic Editor

PLOS One

---

## [Editor Report · Acceptance letter]

PONE-D-25-61177R1

PLOS One

Dear Dr. TANIMURA,

I'm pleased to inform you that your manuscript has been deemed suitable for publication in PLOS One. Congratulations! Your manuscript is now being handed over to our production team.

Kind regards,

on behalf of

Dr. Mohammad Moniruzzaman

Academic Editor

PLOS One